# Microbial Control Agents for Fungus Gnats (Diptera: Sciaridae: *Lycoriella*) Affecting the Production of Oyster Mushrooms, *Pleurotus* spp.

**DOI:** 10.3390/insects12090786

**Published:** 2021-09-03

**Authors:** Valerie M. Anderson, Grace F. H. Sward, Christopher M. Ranger, Michael E. Reding, Luis Canas

**Affiliations:** 1Ohio Agricultural Research and Development Center, Department of Entomology, The Ohio State University, Wooster, OH 44691, USA; anderson.3261@buckeyemail.osu.edu (V.M.A.); sward.6@buckeyemail.osu.edu (G.F.H.S.); canas.4@osu.edu (L.C.); 2USDA-Agricultural Research Service, Horticultural Insects Research Lab, 1680 Madison Ave., Wooster, OH 44691, USA; mike.reding@usda.gov

**Keywords:** *Pleurotus ostreatus*, *Pleurotus columbinus*, *Lycoriella* sp., *Bacillus thuringiensis* var. *israelensis*, *Steinernema feltiae*

## Abstract

**Simple Summary:**

Fungus gnat larvae and adults are challenging insect pests affecting the production of oyster mushrooms (*Pleurotus* spp.). The objectives of this study were to develop a container bioassay and assess the bacteria *Bacillus thuringiensis* var. *israelensis* (*Bti*) and entomopathogenic nematode *Steinernema feltiae* as microbial control agents against fungus gnats. We hypothesized that fungus gnat larvae and the subsequent emergence of adults would be reduced by incorporating *Bti* and *S. feltiae* into straw substrate used for producing oyster mushrooms. A series of bioassays were conducted using straw inoculated with *Pleurotus columbinus* and *Pleurotus ostreatus*. Fewer fungus gnats emerged from substrate treated with *Bti* compared to *S. feltiae* and distilled water. *Steinernema feltiae* was generally ineffective. Incorporating *Bti* and *S. feltiae* into the straw substrate did not appear to impede colonization by *P. ostreatus*. The current study indicates that *Bti* could be useful as a sustainable pest management technique for producing oyster mushrooms.

**Abstract:**

Infestations of fungus gnats (Diptera: Sciaridae) can reduce the production of oyster mushrooms (*Pleurotus* spp.) grown as food crops within controlled environments. The objectives of this study were to assess the efficacy of *Bacillus thuringiensis* var. *israelensis* (*Bti*) and *Steinernema feltiae* against fungus gnat larvae. A bioassay was developed, whereby pasteurized straw was inoculated with *Pleurotus columbinus* and treated with *Bti* (Gnatrol^®^), *S. feltiae* (Nemashield^®^), or water. Fungus gnats (*Lycoriella* sp.) were released into each bioassay container for ovipositing onto the straw, thereby exposing the F_1_ larvae to treated or untreated substrate. Sticky cards within the containers entrapped fungus gnats emerging from the substrate as an indicator of larval survivorship. Following three bioassays, fewer fungus gnats emerged from straw treated with *Bti* compared to *S. feltiae* and the water control. Three additional bioassays using *Pleurotus ostreatus* also demonstrated that fewer fungus gnats emerged from straw treated with *Bti* compared to *S. feltiae* and the untreated control. *Steinernema feltiae* was generally ineffective. Monitoring substrate weight in the bioassay containers over time indicated that *Bti* and *S. feltiae* did not impede colonization by *P. ostreatus*. Incorporating *Bti* into straw substrate is a promising approach for managing fungus gnats infesting *Pleurotus* spp.

## 1. Introduction

Oyster mushrooms, *Pleurotus* spp., are naturally occurring throughout temperate and tropical regions of the world [1]. As saprophytic fungi, *Pleurotus* spp. facilitate wood decomposition, especially dying or dead hardwoods, and are critical for breaking down difficult-to-digest components such as cellulose into useable nutrients for the environment [2]. The degradative activity of *Pleurotus* spp. combined with other fungi and bacteria subsequently benefits the soil community and facilitates the formation of fertile soil [3]. 

*Pleurotus* spp. are cultivated within controlled environments due to their flavor, medicinal properties, vigorous growth, and undemanding cultivation requirements [4]. The U.S. ranks second to China in the production of edible mushrooms [5]. During 2019–2020, the volume of sales of the mushroom crop in the US totaled 370 million kg of crop valued at USD 1.15 billion [6]. *Pleurotus ostreatus* (Jacq. ex. fr) Kummer. is the second-most cultivated edible mushroom in the U.S. and worldwide after button mushrooms, *Agaricus bisporus* (Lange) Imbach. [5]. *Pleurotus columbinus* Quel. is an additional cultivated species of oyster mushroom [7,8]. Oyster mushrooms are excellent sources of protein, vitamins, minerals, and amino acids, along with possessing strong anti-inflammatory properties [9,10,11]. Cultivating oyster mushrooms is less expensive than other wood-loving mushrooms because the growing substrate can be pasteurized instead of sterilized [5]. A variety of agricultural, forest, and food-processing waste products can be used as substrates for oyster mushrooms [12]. The spent mushroom substrate can serve as feed for livestock and mushroom growth improves the digestibility of wheat straw [13]. There is also a comparatively short time from inoculation to fruiting compared to other edible mushrooms [5]. 

Like other mushroom crops grown in controlled environments, producers of oyster mushrooms are challenged by dipteran pests, including sciarid flies (Diptera: Sciaridae) (Figure 1), phorid flies (Diptera: Phoridae), and cecid flies (Diptera: Cecidomyiidae) [14,15]. Yields of oyster mushrooms are reduced by fly larvae feeding on developing mushroom primordia [16]. Larval and adult fungus gnats can also spread mushroom green mold disease, *Trichoderma* spp., by way of carrying spores on their bodies [17,18,19]. Furthermore, dead fungus gnats tend to adhere to oyster mushrooms as a contaminant increasing processing time by producers prior to shipment. Conventional insecticides are used on a global scale for controlling dipteran pests of oyster mushrooms [15]. However, applications of synthetic insecticides can reduce yields by damaging mushroom sporophores, lead to insecticide resistance development, and pose environmental and health concerns [15,20]. Selling locally grown gourmet mushrooms to restaurants and local markets also requires growers to meet increasing consumer demand for pesticide-free products. 

The bacteria *Bacillus thuringiensis* Berliner (*Bt*) is widely used for the microbial control of insect pests [21]. The mode of action of *Bt* involves a crystalline inclusion (or toxic crystal) that binds to the midgut plasma membrane, resulting in pores that disrupt osmotic balance [21,22]. The strain *Bacillus thuringiensis* var. *israelensis* de Barjac (*Bti*) is particularly toxic to Diptera, especially mosquitoes (Diptera: Culicidae) and several other dipterans [22]. Toxicity of *Bti* can occur within minutes after ingestion by mosquito larvae [22]. Previous studies found promising efficacy of *Bti* at reducing populations of dipteran pests attacking *A. bisporus* [16,23,24]. For instance, incorporating *Bti* ABG-6193 into a manure-based substrate reduced adult emergence of *Lycoriela ingenua* (Dufour) (syn. *L. mali* (Fitch) (Diptera: Sciaridae) and *Megaselia halterata* (Wood) (Diptera: Phoridae) affecting the production of *A. bisporus* mushrooms [16,23]. A review of existing literature did not provide data about the efficacy of *Bti* against dipteran insects infesting substrates used to grow oyster mushrooms, but Wang et al. [8]. reported that various strains of *Bt* did not inhibit hyphal growth of *P. ostreatus* and *Pleurotus geesteranus* Singer. 

Entomopathogenic nematodes in the families Steinernematidae and Heterorhabditidae have received considerable attention as biological control agents of insect pests [25]. Infective juveniles enter host insects through natural openings after which cells of their bacterial symbiont (*Xenorhabdus* spp.) are released into the hemolymph, resulting in death of the host by septicemia within 48 h [25]. *Steinernema feltiae* (Filipjev) is effective against various fungus gnat species (Sciaridae), including species infesting *A. bisporus* production facilities [26]. Several studies have demonstrated that treatment of compost with *S. feltiae* suppresses the emergence of dipteran pests impacting *A. bisporus* production [23,24,27,28], but other studies found limited or negligible efficacy [29,30]. To date, few studies have assessed the efficacy of *S. feltiae* for controlling dipteran pests infesting substrates used to grow oyster mushrooms. 

To establish a basis for sustainable control tactics of fungus gnats affecting the production of *Pleurotus* spp. oyster mushrooms, the objectives of our current study were to: (1) develop a small-scale container bioassay using straw substrate inoculated with *Pleurotus* spp. to promote the life cycle of *Lycoriella* sp. fungus gnats, (2) evaluate the impact of treating straw substrate with *Bti* and *S. feltiae* on the emergence of a *Lycoriella* sp. fungus gnat from straw substrate colonized by *P. columbinus* and *P. ostreatus*, and (3) assess if treating substrate with *Bti* or *S. feltiae* impaired mycelial growth of *P. columbinus* and *P. ostreatus*.

## 2. Materials and Methods

### 2.1. Oyster Mushroom Cultures

Syringes containing spores of *P. columbinus* and *P. ostreatus* suspended in water (Out-Grow Inc., McConnell, IL, USA) were used to inoculate sterilized rye berries within glass canning jars (946 mL) or polypropylene grow bags (61 cm × 24.5 cm, l × w; 0.2 μm filter patch) (Figure 2A–C). Spore syringes were obtained in April 2018 and April 2020 for *P. columbinus* and *P. ostreatus*, respectively. Rye berries were rinsed five times and soaked in water for 24 h, then boiled for 10 min and air dried for 30 min before being sterilized in an autoclave for 120 min at 15 psi. Cultures were incubated at 21 °C for 2–3 weeks to allow for full colonization of the rye berry substrate before using in bioassays. 

### 2.2. Container Bioassays

Sealable plastic containers (13 cm × 8 cm, diam. × height; volume = 1061.9 cm^3^) were used for conducting bioassays (Figure 2A–C). To allow for air exchange, two holes (0.75 cm diameter) were drilled into opposite sides of the containers about 3 cm from the top. A third hole (0.75 cm) was drilled into the bottom for drainage during incubation. A piece of fine mesh fabric (9 cm^2^; 30 × 30 squares per cm^2^) was taped over all the holes to prevent the fungus gnat adults from escaping. Containers were cleaned with liquid dish soap and 70% ethanol prior to using them in two separate bioassays. 

### 2.3. Straw Pasteurization 

Bioassays to assess the efficacy of *Bti* and *S. feltiae* for reducing populations of fungus gnat larvae were first conducted using *P. columbinus* followed by *P. ostreatus*. Pre-cut and bagged straw (EZ-Straw Seeding Mulch^®^) was obtained from Rural King Supply in Wooster, Ohio. Straw substrate was pasteurized on 30 May 2018 for bioassay 1, 14 June 2018 for bioassay 2, and 26 July 2018 for bioassay 3 prior to inoculating with *P. columbinus*. Similarly, straw substrate was pasteurized on 22 June 2020 for bioassay 4, 6 July 2020 for bioassay 5, and 13 July 2020 for bioassay 6 prior to inoculating with *P. ostreatus*. Pre-cut straw was first soaked in warm water with liquid dish soap for 2 h. The straw was drained, rinsed multiple times, transferred into a pillowcase, and then sealed with a nylon cable tie. The pillowcase was fully submerged in water at 65 ± 5 °C for 1 h. The pasteurized straw was drained and air dried until only a few drops of water were produced when squeezing the straw. About 500 mL of pasteurized straw was placed within each bioassay container and then sealed with the container lid.

### 2.4. Bioassays with P. columbinus 

Three bioassays were conducted over time with *P. columbinus*. Pasteurized straw from an individual container was spread within a laminar flow hood on a tray sterilized with 70% ethanol. About 100 mL of rye berries colonized by *P. columbinus* mycelium was then mixed with the straw for inoculation purposes and promptly placed back into the bioassay container. For the three separate bioassays, straw was treated with *Bti* and *S. feltiae* on 31 May 2018, 15 June 2018, and 26 July 2018. For each bioassay, 30 mL of *Bti* suspension in water (1.95 g/L; ABG-6193; Gnatrol^®^; Valent USA Corp., Walnut Creek, CA, USA) or *S. feltiae* suspension in water (6.6 million nematodes/L; NemaShield^®^; BioWorks, Inc., Victor, New York, NY, USA) were separately pipetted onto straw in individual containers (Figure 2B). The same volume of water was applied as an untreated control. After treating, containers used in all three bioassays were held overnight in an incubator at 21 °C prior to infesting with fungus gnats the following day. Each treatment was replicated 10 times during each bioassay. Fresh batches of *S. feltiae* were received from the supplier 2‒7 d prior to use in each bioassay and stored in a refrigerator at 5 °C according to the supplier’s instruction prior to use in bioassays. Viability of *S. feltiae* was confirmed before each bioassay by examining an aliquot of the suspension under a stereomicroscope; viability of an *S. feltiae* suspension was also confirmed by documenting 100% larval mortality of *Galleria mellonella* Linnaeus (data not included). 

Straw substrate inoculated with *P. columbinus* and treated with either *Bti* or *S. feltiae* was then infested one day after treatment for all three bioassays, namely, 1 June 2018, 16 June 2018, and 27 July 2018. An aspirator was used to collect female fungus gnats (*Lycoriella* sp.) from a commercial oyster mushroom grower (Medina, OH, USA) the same day that container infestations occurred (Figure 1). Large females with robust abdomens that were purportedly gravid were distinguished from the smaller males. Five female gnats were collected per vial, then placed on gel ice packs within a cooler and transferred back to the laboratory. One vial of fungus gnats was used to infest each container about 1 h after their collection. Infested containers were held in an incubator at 21 °C. After 21 days, yellow sticky cards (5 cm × 5 cm) were affixed to the inside lid of each container to entrap fungus gnats emerging from the substrate (Figure 2C). Sticky cards were not affixed prior to this duration to avoid moisture accumulating on the adhesive. At 24 days after infestation, fungus gnats were counted on the sticky cards, along with any dead specimens adhering to the mycelium (Figure 2B,C). Voucher specimens of the fungus gnat (*Lycoriella* sp.) used in our study were submitted to the C.A. Triplehorn Insect Collection, The Ohio State University, OSUC consisting of 10 females in ethanol (log # 834901), 10 males in ethanol (log # 834902), 10 females dry mounted on points (log #s 834903–834912), and 10 males dry mounted on points (log #s 834913–834922). 

### 2.5. Bioassays with P. ostreatus

Three additional bioassays were conducted over time with *P. ostreatus*. The aforementioned methods were used, except the pasteurized straw inoculated with *P. ostreatus* was infested with fungus gnats prior to treating with *Bti* and *S. feltiae* (Figure 2B,C). These changes were implemented because treatment with *Bti* and *S. feltiae* prior to infesting could have influenced oviposition behavior by the fungus gnats. Bioassay containers were infested with gravid fungus gnats the day after inoculating with *P. ostreatus* and prior to treating with *Bti* and *S. feltiae*. Bioassay containers enclosing straw inoculated with *P. ostreatus* were infested with five gravid female fungus gnats (*Lycoriella* sp.) from a commercial oyster mushroom facility on 23 June 2020, 07 July 2020, and 14 July 2020 corresponding to three separate bioassays (Figure 2B). Infested containers were held in an incubator at 21 °C.

Treatments for bioassays involving *P. ostreatus* were applied 3 days after infesting with the gravid fungus gnats. This ensured the fungus gnats had time to oviposit onto the inoculated straw substrate. The aforementioned rates and application techniques were used. Each treatment was replicated 10 times during each bioassay. Containers used in all three bioassays with *P. ostreatus* were held in an incubator at 21 °C for 24 days, allowing a full fungus gnat life cycle to elapse. Adult emergence was quantified as previously described. 

### 2.6. Influence of Treatment on Mycelial Growth

Two bioassays were conducted to assess if treatment of the inoculated straw with *Bti* or *S. feltiae* impaired mycelial growth. During both bioassays, pasteurized straw was first inoculated with *P. ostreatus* as previously described, and all containers were then weighed. Bioassay containers were then randomly assigned to treatment with solutions of *Bti* (n = 10), *S. feltiae* (n = 10), or water (n = 10) as previously described, and then weighed on the same day as treatment followed by once a week for three weeks. Individual weights of each empty container were subtracted from the total weight to determine weight of the substrate. Containers were held in an incubator at 21 °C.

### 2.7. Statistics

To assess if treating the substrate with *Bti*, *S. feltiae*, or water influenced emergence of adult fungus gnats, one-way ANOVA and Fisher’s least significant difference (LSD) were used to compare among treatment means within each of the three separate bioassays involving *P. columbinus* and *P. ostreatus* (SAS Institute Inc., Cary, NC, USA). To assess within-treatment variability in fungus gnat emergence, one-way ANOVA and Fisher’s LSD were used to separately compare fungus gnat emergence across the three bioassays with *P. columbinus* and *P. ostreatus* for straw substrate treated with *Bti*, *S. feltiae*, or water. Count data were square-root transformed prior to analysis, but untransformed data are presented. To assess if treating the substrate with *Bti*, *S. feltiae*, or water influenced mycelial growth, a repeated measures ANOVA was used to test for between-treatment effects and within-treatment effects across the five time points at which substrate weights were measured. To test for a within-subject time effect, substrate weights among all treatments within a bioassay were pooled and substrate weights across the five time points were compared using a one-way ANOVA and Fisher’s LSD. 

## 3. Results

Pasteurized straw placed within the bioassay containers was thoroughly colonized by *P. columbinus* and *P. ostreatus*. White mycelium of the *Pleurotus* spp. was observed growing on the straw substrate 4−5 days after inoculation (Figure 2A–C), and the straw within each container was often fully colonized within 14 days after inoculation (Figure 2A–C). The fungal growth supported development of *Lycoriella* sp. fungus gnats (Figure 2A–C), which allowed for bioassays to evaluate the efficacy of *Bti* and *S. feltiae* against the larvae. 

### 3.1. Bioassays with P. columbinus

Significantly fewer fungus gnat adults emerged from straw treated with *Bti* compared to *S. feltiae* and water during the first bioassay using substrate inoculated with *P. columbinus* (Figure 3A: *F*_2, 26_ = 3.82; *p* = 0.04). The *Bti* treatment resulted in a 40% reduction in adult fungus gnats compared to the water control versus 8% for *S. feltiae*. A significant difference was not detected in fungus gnat adults that emerged from straw treated with *S. feltiae* compared to water. 

In the second bioassay with *P. columbinus*, significantly fewer adults emerged from straw treated with *Bti* compared to the *S. feltiae* treatment and water control (Figure 3B: *F*_2, 27_ = 7.66; *p* = 0.002). A significant difference was not detected in the number of adults that emerged from straw treated with *S. feltiae* compared to the water control. The *Bti* treatment resulted in a 57% reduction in adult emergence compared to the water control versus 27% for *S. feltiae*. 

Significantly fewer adults emerged from straw treated with *Bti* compared to *S. feltiae* and water control during the third bioassay with *P. columbinus* (Figure 3C: *F*_2, 27_ = 5.18; *p* = 0.012). The *Bti* treatment resulted in a 51% reduction in adult emergence compared the water control versus 2% for *S. feltiae*. A significant difference in adult emergence was not detected between straw treated with *S. feltiae* and the water control. 

During the bioassays with *P. columbinus*, a significant difference was not detected in emergence of fungus gnats from the water-treated control substrate among the three bioassays (Figure 3A–C: *F*_2, 27_ = 3.16; *p* = 0.06). A significant difference was also not detected in emergence of fungus gnats from the *S. feltiae* treated substrate among the three bioassays (Figure 3A–C: *F*_2, 27_ = 3.35; *p* = 0.44) or the *Bti* treated substrate among the three bioassays (Figure 3A–C: *F*_2, 27_ = 3.35; *p* = 0.89).

### 3.2. Bioassays with P. ostreatus

During the first bioassay using substrate inoculated with *P. ostreatus*, significantly fewer adults emerged from straw treated with *Bti* compared to *S. feltiae* and the water control (Figure 4A: *F*_2, 26_ = 5.16; *p* = 0.013). The *Bti* treatment resulted in a 53% reduction compared to the water control versus a 6.3% increase for *S. feltiae*. A significant difference was not detected in the number of adults that emerged from straw treated with *S. feltiae* vs. the water control. 

In the second bioassay with *P. ostreatus*, significantly fewer fungus gnats emerged from straw treated with *Bti* compared to *S. feltiae* and the water control (Figure 4B: *F*_2, 27_ = 14.35; *p* < 0.0001). Significantly fewer adults also emerged from straw treated with *S. feltiae* compared to the water control. The *Bti* treatment resulted in an 89% reduction compared to the water control versus 57% for *S. feltiae*. 

In the third bioassay with *P. ostreatus*, significantly fewer fungus gnats emerged from the *Bti* compared to the *S. feltiae* treatment and the untreated control (Figure 4C). Significantly fewer fungus gnats also emerged from straw treated with *S. feltiae* compared to the untreated control (Figure 4C: *F*_2, 27_ = 24.53; *p* < 0.0001). The *Bti* treatment resulted in an 88% reduction and *S. feltiae* treatment resulted in an 53% reduction in fungus gnat emergence compared to the water control.

Within-treatment variability in fungus gnat emergence was documented among the three bioassays with *P. ostreatus*. Significantly more adults emerged during the second bioassay compared to the first and third bioassays with *Bti* treated substrate (Figure 4A–C: *F*_2, 27_ = 5.79; *p* = 0.008). Significantly more adults emerged from substrate treated with *S. feltiae* during the second bioassay compared to the first and third bioassays (Figure 4A–C: *F*_2, 27_ = 25.39; *p* < 0.0001). Significantly more adults emerged during the second bioassay compared to the first and third bioassays from the water-treated control substrate (Figure 4A–C: *F*_2, 27_ = 9.30; *p* = 0.001). 

### 3.3. Influence of Treatment on Mycelial Growth

Recording substrate weight over time following inoculation with *P. ostreatus* and treatment with *Bti* and *S. feltiae* during two separate bioassays allowed for assessing if the treatments impaired mycelial growth. During both bioassays, a significant between-subjects effect was not detected among treatments over time (between-subjects, repeated measures ANOVA; Figure 5A: *F*_2, 27_ = 1.85; *p* = 0.18; Figure 5B: *F*_2, 27_ = 0.09; *p* = 0.92). Thus, at each time point, a difference was not detected in weight of substrates treated with *Bti*, *S. feltiae*, or water during these two bioassays (Figure 5A,B). 

A significant time effect was detected in the weight of substrates treated with *Bti*, *S. feltiae*, and water (within-subjects, repeated measures ANOVA; Figure 5A: *F*_4, 108_ = 1308.9; *p* < 0.0001; Figure 5B: *F*_4, 108_ = 3335.94; *p* < 0.0001). During these bioassays, substrate weight initially increased after treating with solutions containing *Bti*, *S. feltiae*, or water alone compared to the day when the substrate was first inoculated. Substrate weight then consistently decreased over time during both bioassays such that substrate weight 3 weeks after treatment was significantly less compared to the day of treatment with *Bti*, *S. feltiae*, and water alone (Figure 5A: *F*_4, 145_ = 41.64; *p* < 0.0001; Figure 5B: *F*_4, 145_ = 48.26; *p* < 0.0001).

## 4. Discussion

Increased production of oyster mushrooms on a global scale warrants the evaluation of biocontrol agents as alternatives to conventional insecticides for controlling fungus gnat pests. Our current study established a small-scale bioassay technique for evaluating *Bti* and *S. feltiae*, but the technique could also be used to evaluate insect growth regulators (IGRs) [15] and botanically based extracts [31,32] incorporated into the growing substrate. Using this bioassay technique, our current study demonstrated that incorporating *Bti* (ABG-6193; Gnatrol^®^) into a straw mushroom growing substrate for *P. columbinus* and *P. ostreatus* reduced the emergence of adult fungus gnats. We hypothesize that the reduction in adult emergence was a function of larval mortality following exposure to *Bti* within the treated substrates. In contrast, *S. feltiae* ranged from ineffective to slightly effective at reducing adult emergence, which could be a function of *Pleurotus* spp. being known to immobilize and digest nematodes [33]. These results will assist with further development of an IPM program to control dipteran pests of oyster mushrooms, which is a function of four key principles—exclusion, monitoring, sanitation, and pest control [14].

To date, most studies have tested *Bt* formulations for protecting *A. bisporus* crops against dipteran insects. For instance, Keil [16] reported that *Bti* ABG-6193 reduced populations of *L. ingenua* and *Megaselia halterata* (Wood) (Diptera: Phoridae) affecting the production of *A. bisporus* mushrooms. Yield benefits were also documented, which could be due to improved larval control, along with proteins and lipids associated with *Bti* supplementing the nutritional requirements of *A. bisporus* [16]. Erler et al. [23] also found *Bti* ABG-6193 reduced damage by larvae of *M. halterata* to *A. bisporus*, and there were no differences in adult emergence compared to substrate treated with chlorpyrifos. In contrast, Jess and Kilpatrick [29] reported *Bti* (Skeetal^®^ Novo, Bagsværd, Denmark) did not reduce populations of *Lycoriella solani* (Diptera: Sciaridae) emerging from compost used to grow *A. bisporus*, nor benefit yield. Our current study indicates *Bti* could be useful for the protection and production of oyster mushroom crops. An extensive body of research conducted over decades continues to demonstrate the use of *Bt* for insect pest control in food crops is safe for vertebrates [34].

Studies have documented that treating compost with entomopathogenic nematodes reduces the emergence of fungus gnats affecting the production of *A. bisporus*, but variability in efficacy has been reported [23,24,27,28,35,36,37]. For instance, Jess and Kilpatrick [29] demonstrated that application of *S. feltiae* to compost during the colonization phase by *A. bisporus* did not immediately reduce sciarid emergence and only resulted in a limited reduction at later crop stages. Yet, an application of *S. feltiae* immediately after casing of the colonized compost reduced emergence of fungus gnats by 82%. Navarro and Gea [30] found that treating cased compost with *S. feltiae* and *S. carpocapsae* reduced the emergence of *Lycoriella auripila* Winnertz (Diptera: Sciaridae) but not *Megaselia halterata* (Wood) (Diptera: Phoridae). Since our study only tested *S. feltiae*, additional studies are warranted to test *S. carpocapsae* against fungus gnats infesting oyster mushroom substrates. 

Overall, our current study found that straw substrate inoculated with *P. columbinus* and *P. ostreatus* and treated with *S. feltiae* had little to no effect on the emergence of fungus gnats. Efficacy of *S. feltiae* might have been improved by delaying application until larval hosts were certain to be available within the substrate. The carnivorous activity of *P. ostreatus* and *P. columbinus* against *S. feltiae* could also have resulted in reduced efficacy. Although we did not observe this activity, it is reported in the literature. More than 150 species of fungi are known to attack nematodes, including *Pleurotus* spp. [38,39,40]. Hyphae of *P. ostreatus* produce secretions containing trans-2-decenedioic acid that allow the carnivorous fungus to quickly immobilize and digest nematodes [33]. Similarly, fatty acids secreted by *Pleurotus djamor* (Fr.) Boedijn are also lethal to the parasitic nematode *Haemonchus contortus* Rud. [41]. Marlin et al. [42] documented examples of susceptibility and resistance among thirteen bacterial-feeding nematode species to toxin-producing isolates of *Pleurotus pulmonarius* (Fr.) Quél. and *P. ostreatus*. Additional studies are thereby warranted to assess the susceptibility of entomopathogenic nematodes in the families Steinemernatidae and Heterorhabditidae to nematophagous *Pleurotus* spp. For instance, recent studies demonstrated that aegerolysin proteins produced by *P. ostreatus* bind with insect midguts and create pores that disrupt osmosis and result in insect death [43,44].

Variability in adult emergence was documented among the three bioassays with *P. ostreatus*. This variability might be attributed to differences in the age and fecundity of field-collected individuals used in our bioassays as opposed to using lab-reared individuals of a known age and reproductive status. A rearing protocol for this *Lycoriella* sp. has since been established using straw substrate inoculated with *P. columbinus* that will provide lab-reared specimens for future studies (Sward, unpub. data). 

A study by Wang et al. [8] reported that several strains of *Bt* did not inhibit hyphal growth of *P. ostreatus* and *P. geesteranus*. Our current study also found no detectable inhibition of growth by treating straw substrate with *Bti* and *S. feltiae*. Substrate weights among the three treatments remained the same over time, but the substrate weight within each treatment changed as expected. For instance, the documented weight increase immediately after treatment can be attributed to water being added to the substrate. Afterwards, the substrate weight steadily decreased over time across all treatments. The decrease in weight can be attributed to the mycelium of *P. ostreatus* and *P. columbinus* degrading and consuming the straw substrate over time, thereby decreasing the overall container weight. 

A cost-benefit analysis would be useful to compare *Bti* with other control tactics against fungus gnats infesting oyster mushroom substrates. Additional studies are also warranted to compare varying rates of *Bti* for incorporating into substrates used for growing *Pleurotus* spp. oyster mushrooms, along with impacts on yield. Keil [16] documented increased yields of *A. bisporus* following treatment of compost with *Bti* ABG-6193, which could have been due to suppression of dipteran pests and/or proteins and lipids within the *Bti* formulation providing nutritional supplement. Results from our current study indicate *Bti* could be a promising control tactic for growers, but a plan would also need to be implemented to counteract the selection of insects resistant to the insecticidal proteins. 

## Figures and Tables

**Figure 1 insects-12-00786-f001:**
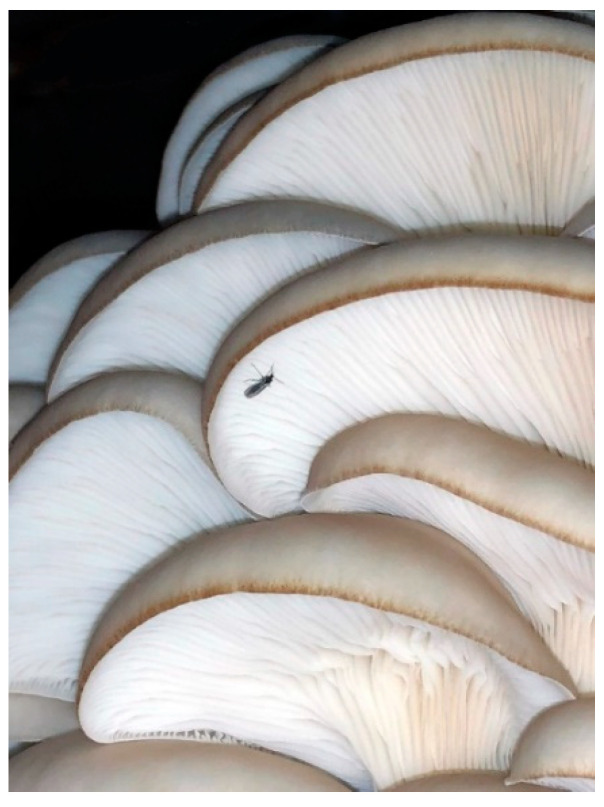
Adult fungus gnat, *Lycoriella* sp., resting on the gills of an oyster mushroom, *P. ostreatus*.

**Figure 2 insects-12-00786-f002:**
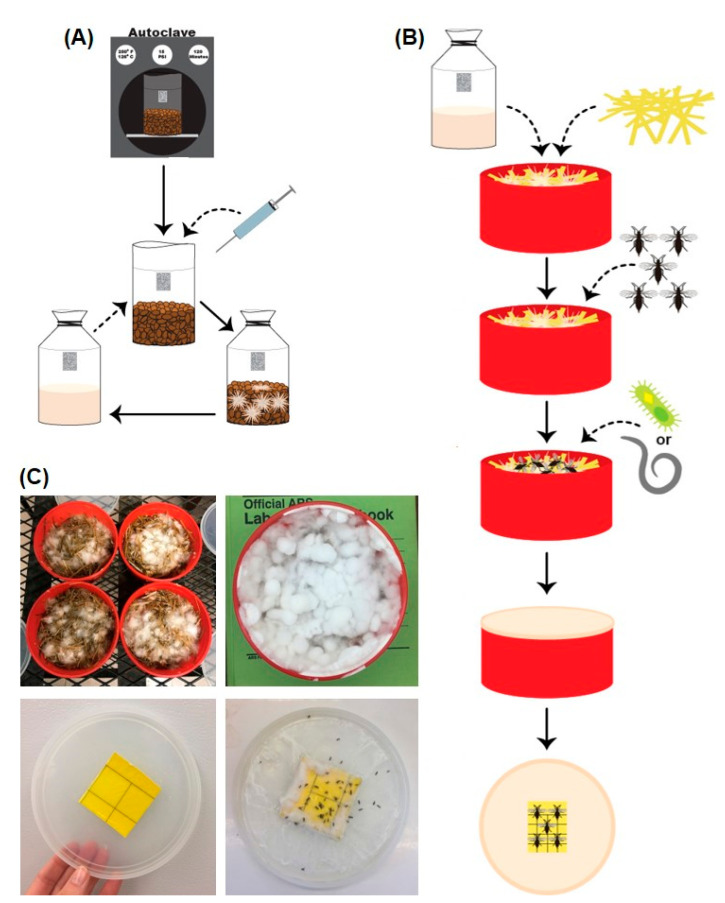
Infographic demonstrating the bioassay methods used to test the effect of incorporating *Bacillus thuringiensis* var. *israelensis* and *Steinernema feltiae* into straw substrate inoculated with *Pleurotus* spp. on the emergence of fungus gnats (*Lycoriella* sp.). Solid lines indicate processes and dashed lines indicate added elements. (**A**) The process of preparing rye berry ‘spawn’ consisted of substrate sterilization, inoculation with a *Pleurotus* spp. spore syringe, and incubation for mycelium colonization. Two spawn bags were prepared to inoculate all 30 substrate containers within each bioassay. (**B**) The process of preparing the bioassay containers consisted of mixing colonized rye berries and pasteurized straw in bioassay containers, infesting with fungus gnats, incorporating biological control agents (*Bti* or *S. feltiae*), and entrapping emerged adults. Ten substrate containers per treatment were prepared for each bioassay for a total of 30 containers. (**C**) Bioassay containers with inoculated straw and sticky cards affixed to the lids with entrapped adult fungus gnats.

**Figure 3 insects-12-00786-f003:**
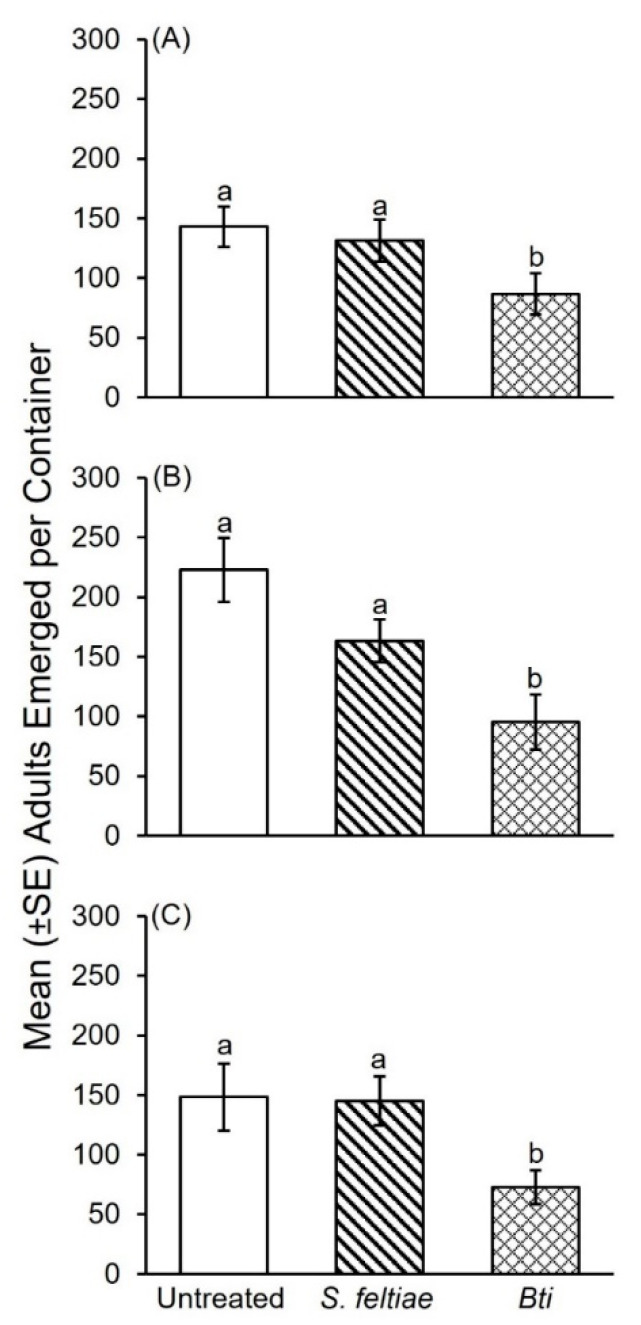
(**A**–**C**) Mean (± SE) number of adult fungus gnats that emerged from straw substrate inoculated with *P. columbinus* and treated with either a water control, *S. feltiae* (Nemashield^®^), or *B. thuringiensis* var. *israelensis* (ABG-6193; Gnatrol^®^) during three independent bioassays. Three independent bioassays were conducted over time and presented as A–C. Means within a bioassay with different letters are significantly different (one-way ANOVA; Fisher’s LSD test; α = 0.05).

**Figure 4 insects-12-00786-f004:**
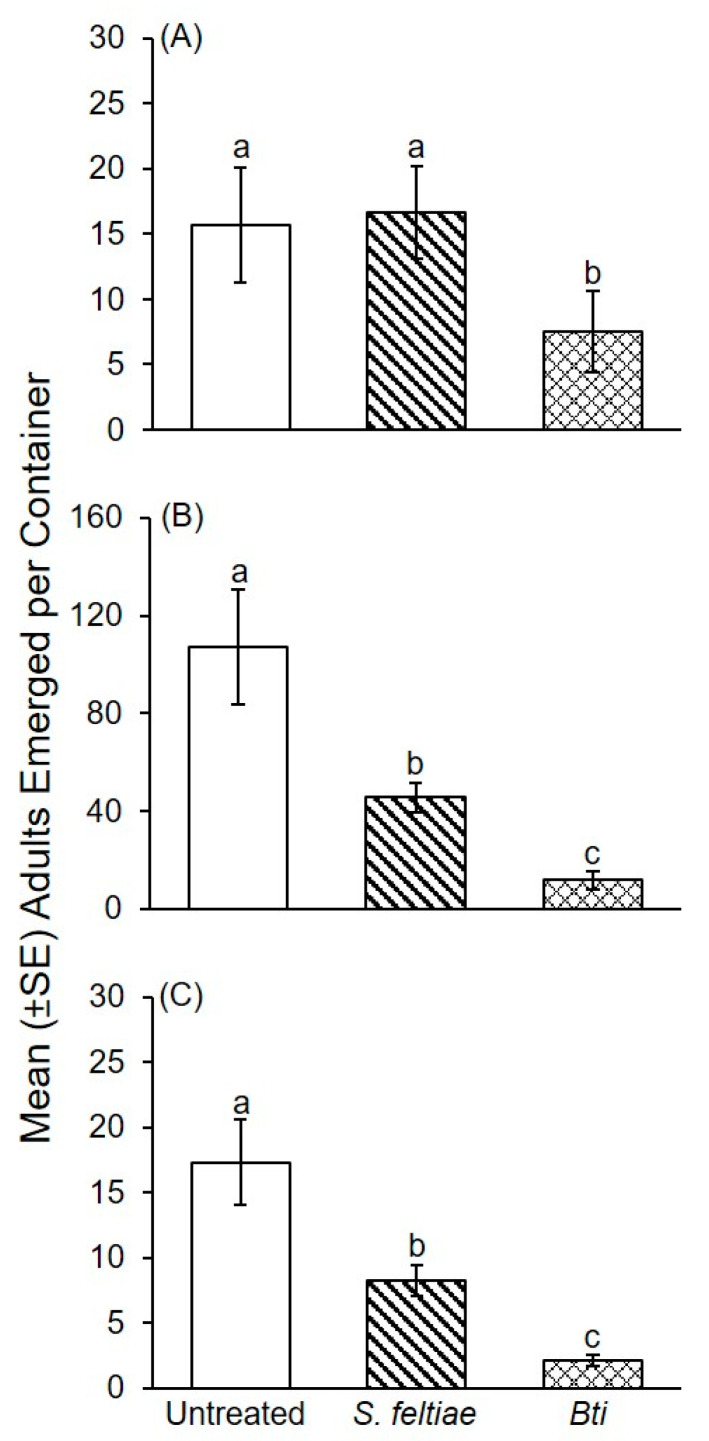
(**A**–**C**) Mean (± SE) number of adult fungus gnats that emerged from straw substrate inoculated with *P. ostreatus* and treated with either a water control, *S. feltiae* (Nemashield^®^), or *B. thuringiensis* var. *israelensis* (ABG-6193; Gnatrol^®^) during three independent bioassays. Three independent bioassays were conducted over time and presented as A-C. Means within a bioassay with different letters are significantly different (one-way ANOVA; Fisher’s LSD test; α = 0.05).

**Figure 5 insects-12-00786-f005:**
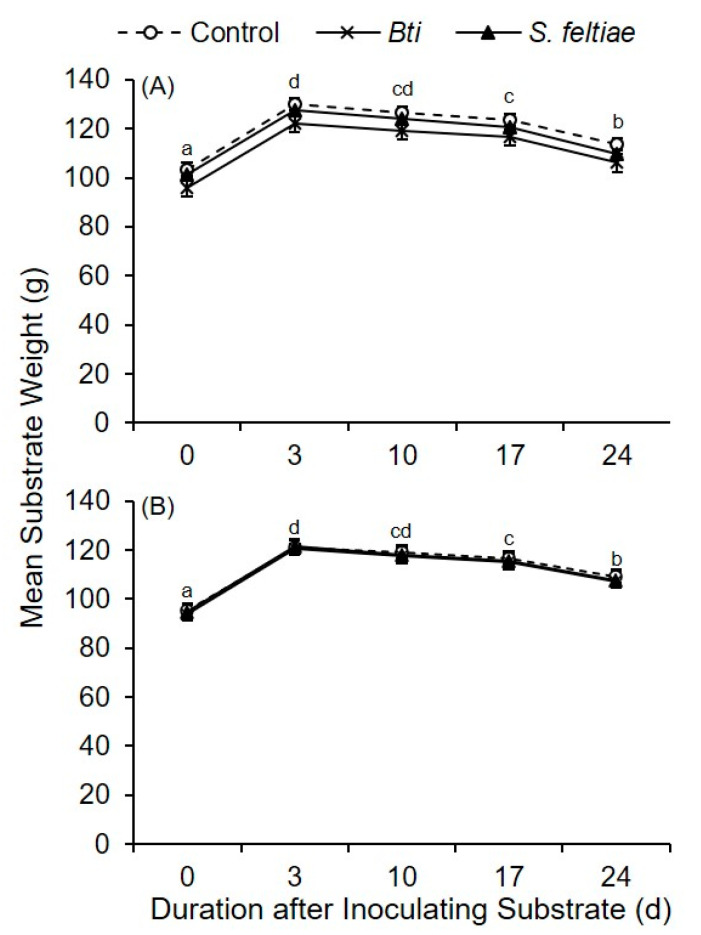
(**A**,**B**) Mean (±SE) weight of straw in bioassay containers inoculated with *P. ostreatus* and treated at 3 d post-inoculation with *Bti*, *S. feltiae*, or water as an untreated control. Significant differences were not detected in container weights among treatments over time (between-subjects, repeated measures ANOVA, α = 0.05), but significant within-treatment effects were detected over time (within-subjects, repeated measures ANOVA, α = 0.05). A significant difference was detected in substrate weights across the five time points when pooled among treatments; different letters indicate significant differences across the five time points within each bioassay (one-way ANOVA and LSD, α = 0.05).

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
