# Peer review of "Microbial Control Agents for Fungus Gnats (Diptera: Sciaridae: Lycoriella) Affecting the Production of Oyster Mushrooms, Pleurotus spp."

_insects, 2021, doi:10.3390/insects12090786_

Round 1

Reviewer 1 Report

Authors have presented a straightforward, generally well-written paper about the possibility of using Bti or a nematode for control of fungus gnats in mushroom systems.  The following points should be addressed:

Title and elsewhere:  The nematode Steinernema feltiae is typically not considered a microbe.  Referring to it as a “microbial control agent” is imprecise, if not incorrect.

Lines 177 and 178:  Please check for proper subject–verb use in both cases: “was placed” and  “were then sealed”

Line 206:  How did authors determine that the female fungus gnats were gravid?  Were they actually able to see the eggs through the abdominal cuticle?  The experiment depends on an equal number of gravid flies in each container.  Please provide details.

The infographic is a nice example of simplicity and clarity in summarizing the methods.  It would be even more helpful if authors included sample sizes.  Please provide sample sizes, either in the infographic itself or the caption.

Lines 227-228:  “containers containing” is redundant; rephrase.

Lycoriella sp.:  It is unfortunate that authors do not provide the species name (or species names).  Although I understand that taxonomic expertise for sciarids is scarce, the authors could try to address this matter, at the very least by formally depositing vouchers specimens in their institutional collection and stating in the Materials and Methods where the specimens were deposited and the number of each life stage and sex that were deposited. 

By referring to Lycoriella “sp.”, the authors are indicating that only one species of Lycoriella was present.  If they do not know the species, how can they be confident that only one species was present?

Lines 288-289: “compared to the” (not “compared the”)

Lines 303, 305-306, 347, 350, 397-399, 431, and elsewhere:  Recast the sentences to avoid the banal phrase “there was no”

Fig. 5A-B (should be Fig. 5A, B):  In the caption, please indicate that “with water” means untreated.

Line 414:  “presumably by targeting the larvae”:  Why “presumably”?  The mode of action is typically in the guts of larvae, which have the appropriate pH.  Can authors confirm that the larvae were killed by the Bti?

Line 421:  “As previously noted”:  If already noted, then why note it again?  Delete.

Authors noted the concerns for human health if pesticides are used (e.g., lines 89-93). A brief statement about human safety with regard to using Bti in the mushroom industry could be added in the Discussion.

Lines 467-469:  Please give the rationale for suggesting that variability in adult emergence could be attributed to using field-collected individuals for infesting the bioassay containers.

Author Response

Reviewer #1

Authors have presented a straightforward, generally well-written paper about the possibility of using Bti or a nematode for control of fungus gnats in mushroom systems.  The following points should be addressed:

Comment 1-1: Title and elsewhere: The nematode Steinernema feltiae is typically not considered a microbe.  Referring to it as a “microbial control agent” is imprecise, if not incorrect.

Authors’ Response: We appreciate the feedback. Lacey and Shapiro-Ilan (2008) Annu Rev Entomol considered microbial control agents (MCAs) to include viruses, Bacillus thuringiensis, fungi, and entomopathogenic nematodes (EPNs). Nematodes were included as microbial control agents since bacteria associated with the nematodes are responsible, in part, for killing the host. We propose to use the current title.

Comment 1-2: Lines 177 and 178: Please check for proper subject–verb use in both cases: “was placed” and “were then sealed”

Authors’ Response: The text was revised to note: “About 500 ml of pasteurized straw was placed within each bioassay container and then sealed with the container lid.”

Comment 1-3: Line 206:  How did authors determine that the female fungus gnats were gravid?  Were they actually able to see the eggs through the abdominal cuticle?  The experiment depends on an equal number of gravid flies in each container.  Please provide details.

Authors’ Response: The text was revised to note: “Large females with robust abdomens that were purportedly gravid were distinguished from the smaller males.”

Comment 1-4: The infographic is a nice example of simplicity and clarity in summarizing the methods.  It would be even more helpful if authors included sample sizes.  Please provide sample sizes, either in the infographic itself or the caption.

Authors’ Response: We appreciate the feedback. The infographic caption was revised to note (lines 379-382): “(A) The process of preparing rye berry ‘spawn’ consisted of substrate sterilization, inoculation with a Pleurotus spp. spore syringe, and incubation for mycelium colonization. Two spawn bags were prepared to inoculate all 30 substrate containers within each bioassay.” The infographic caption was also revised to note: “(B) The process of preparing the bioassay containers consisted of mixing colonized rye berries and pasteurized straw in bioassay containers, infesting with fungus gnats, incorporating biological control agents (Bti or S. feltiae), and entrapping emerged adults. Ten substrate containers per treatment were prepared for each bioassay for a total of 30 containers.”

Comment 1-5: Lines 227-228: “containers containing” is redundant; rephrase.

Authors’ Response: The sentence was revised to note: “Bioassay containers enclosing straw inoculated with P. ostreatus were infested with five gravid female fungus gnats (Lycoriella sp.) from a commercial oyster mushroom facility on 23-June-2020, 07-July-2020, and 14-July-2020 corresponding to three separate bioassays (Figure 2B).

Comment 1-6: Lycoriella sp.: It is unfortunate that authors do not provide the species name (or species names).  Although I understand that taxonomic expertise for sciarids is scarce, the authors could try to address this matter, at the very least by formally depositing vouchers specimens in their institutional collection and stating in the Materials and Methods where the specimens were deposited and the number of each life stage and sex that were deposited. 

Authors’ Response: We appreciate the feedback. Specimens were sent to the USDA Systematics Entomology Lab, but only the genus Lycoriella sp. was provided. The text was revised to note: “Voucher specimens of the fungus gnat (Lycoriella sp.) used in our study (20 females and 20 males) were submitted to the Museum of Biological Diversity, The Ohio State University.”

Comment 1-6: By referring to Lycoriella “sp.”, the authors are indicating that only one species of Lycoriella was present.  If they do not know the species, how can they be confident that only one species was present?

Authors’ Response: We agree it is possible that more than once species might have been collected, but examinations of specimen morphology (and our past experience with rearing and conducting experiments with fungus gnats) indicated one species was used in the experiments.

Comment 1-7: Lines 288-289: “compared to the” (not “compared the”)

Authors’ Response: The sentence was revised to note: “The Bti treatment resulted in a 40% reduction in adult fungus gnats compared to the water control versus 8% for S. feltiae.”

Comment 1-8: Lines 303, 305-306, 347, 350, 397-399, 431, and elsewhere:  Recast the sentences to avoid the banal phrase “there was no”

Authors’ Response: The text was revised throughout to omit “there was no”. For instance:

Lines 288-294: “Significantly fewer fungus gnat adults emerged from straw treated with Bti compared to S. feltiae and water during the first bioassay using substrate inoculated with P. columbinus (Figure 3A: F2, 26 = 3.82; p = 0.04). The Bti treatment resulted in a 40% reduction in adult fungus gnats compared to the water control versus 8% for S. feltiae. A significant difference was not detected in fungus gnat adults that emerged from straw treated with S. feltiae compared to water.”

Comment 1-9: Fig. 5A-B (should be Fig. 5A, B):  In the caption, please indicate that “with water” means untreated.

Authors’ Response: The text was revised to note: “Figure 5A,B. Mean weight of straw in bioassay containers inoculated with P. ostreatus and treated at 3 d post-inoculation with Bti, S. feltiae, or water as an untreated control.”

Comment 1-10: Line 414:  “presumably by targeting the larvae”:  Why “presumably”?  The mode of action is typically in the guts of larvae, which have the appropriate pH.  Can authors confirm that the larvae were killed by the Bti?

Authors’ Response: The text was revised to note: “Using this bioassay technique, our current study demonstrated that incorporating Bti (ABG-6193; Gnatrol®ď¸Ź) into a straw mushroom growing substrate for P. columbinus and P. ostreatus reduced the emergence of adult fungus gnats. We hypothesize the reduction in adult emergence was a function of larval mortality following exposure to Bti within the treated substrates.”

Comment 1-11: Line 421:  “As previously noted”:  If already noted, then why note it again?  Delete.

Authors’ Response: The sentence was deleted.

Comment 1-12: Authors noted the concerns for human health if pesticides are used (e.g., lines 89-93). A brief statement about human safety with regard to using Bti in the mushroom industry could be added in the Discussion.

Authors’ Response: The text was revised to note: “An extensive body of research conducted over decades continues to demonstrate the use of Bt for insect pest control in food crops is safe for vertebrates [36].”

Comment 1-13: Lines 467-469: Please give the rationale for suggesting that variability in adult emergence could be attributed to using field-collected individuals for infesting the bioassay containers.

Authors’ Response: The text was revised to note: “Variability in adult emergence was documented among the three bioassays with P. ostreatus. This variability might be attributed to differences in the age and fecundity of field collected individuals used in our bioassays as opposed to using lab reared individuals of a known age and reproductive status. A rearing protocol for this Lycoriella sp. has since been established using straw substrate inoculated with P. columbinus that will provide lab-reared specimens for future studies (Sward, unpub. data).”

Reviewer 2 Report

I think the manuscript is interesting and that some researchers in the field may be interested to take a look on.

I think the most relevant aspect of the article is the conclusion that Bacillus thuringiensis israelensis into straw substrate is a promising approach for managing fungus gnats infesting Pleurotus spp.

I think the manuscript may be published in the present form.

Author Response

Reviewer #2

I think the manuscript is interesting and that some researchers in the field may be interested to take a look on.

I think the most relevant aspect of the article is the conclusion that Bacillus thuringiensis israelensis into straw substrate is a promising approach for managing fungus gnats infesting Pleurotus spp.

I think the manuscript may be published in the present form.

Authors’ Response: We appreciate the supportive comments. 

Reviewer 3 Report

This manuscript reports the efficacy of Bt and EPN on the reduction of fungus gnats on Hiratake, or oyster mushroom. Although the results were not outstanding, they may be useful from the viewpoint of the biology of these organisms. The manuscript may be published with medium to major revision. I am referring to issues that should be considered in the revision of the manuscript below.

No evaluation of economic efficiency

The authors did not evaluate the economic efficiency in using Bt for the insect pests. In this point the authors should avoid referring to the possibility that Bt can be used for the control of the gnats in the mushroom cultivation, however suggestions may be possible.

Use of S. feltiae (L121-130)

It is not clear why the authors examined not S. carpocapsae but S. feltiae. The authors describe positively the efficacy of the former but rather negatively later as a biological control agent. The authors should clearly explain why they selected S. feltiae in their study. I do not think these sentences are necessary in this manuscript.

The author(s)

The authors of species that were used in the study must be referred to (L15, 16, 32, 40).

Not italicize

Bti should not be italicized, since this is neither genus nor species, or even not indicating any genes.

Introduction

It is too long, especially descriptions on the mushroom. It should be reduced in half.

Spores (L144)

When were they obtained? Refer to the date.

Citation order of figures

Figure 2 is cited before Figure 1 (L208). Number the figures as cited.

Figures 1 and 2

These figures may be removed from the manuscript. Without them, the manuscript can be understood by readers.

25 cm2 (L213)

Do you mean 5 by 5 cm? The figure is more important than the size, since the size can be calculated from the figure, which cannot be known from the size though.

L223-225

Do you mean that the first method was not good to evaluate the efficacy of the agents and that the results could have been biased? If so, Bioassay 1 should be removed from the manuscript.

L354-355

Refer to the other df. F-test is determined with two dfs.

Figures 3-4

Explain the three panels in the legend of each figure.

Figure 5

Errors should be indicated to each mean. If the addition of errors makes the figure messy, consider providing a table.

L454

Is it not contradictory with panels B and C in Figure 4?

Author Response

Reviewer #3

This manuscript reports the efficacy of Bt and EPN on the reduction of fungus gnats on Hiratake, or oyster mushroom. Although the results were not outstanding, they may be useful from the viewpoint of the biology of these organisms. The manuscript may be published with medium to major revision. I am referring to issues that should be considered in the revision of the manuscript below.

Authors’ Response: We appreciate the reviewer’s feedback.   

Comment 3-1:

No evaluation of economic efficiency. The authors did not evaluate the economic efficiency in using Bt for the insect pests. In this point the authors should avoid referring to the possibility that Bt can be used for the control of the gnats in the mushroom cultivation, however suggestions may be possible.

Authors’ Response: We appreciate the insight. The goal of our study was to establish an initial baseline of data on microbial control agents against fungus gnats infesting oyster mushroom substrates. We agree that evaluating the economic efficiency of selected treatments is an additional study that warrants examination. The text was revised to note: “A cost-benefit analysis would be useful to compare Bti with other control tactics against fungus gnats infesting oyster mushroom substrates.”

Comment 3-2:

Use of S. feltiae (L121-130). It is not clear why the authors examined not S. carpocapsae but S. feltiae. The authors describe positively the efficacy of the former but rather negatively later as a biological control agent. The authors should clearly explain why they selected S. feltiae in their study. I do not think these sentences are necessary in this manuscript.

Authors’ Response: As noted in the Introduction, most studies conducted to date have evaluated S. feltiae against fungus gnats infesting mushroom substrates. Specifically, it is noted: “Several studies have demonstrated that treatment of compost with S. feltiae suppresses the emergence of dipteran pests impacting A. bisporus production [23-24,27-28], but other studies found limited or negligible efficacy [29-30].” We therefore focused our bioassays on S. feltiae. The text was revised to note: “Since our study only tested S. feltiae, additional studies are warranted to test S. carpocapsae against fungus gnats infesting oyster mushroom substrates.”

We believe the details provided in L121-130 provide useful background information and help to establish our testing of S. feltiae. We defer to the editors about their opinion to reduce the length of the Introduction.

Comment 3-3: The author(s). The authors of species that were used in the study must be referred to (L15, 16, 32, 40).

Authors’ Response:

The text was revised throughout (except for the simple summary) to include the authors of all species upon first mention.

Comment 3-4:

Not italicize. Bti should not be italicized, since this is neither genus nor species, or even not indicating any genes.

Authors’ Response:Since Bti is an abbreviation for Bacillus thuringiensis israelensis, many publications italicize Bti. We will defer to the editors for their opinion on not italicizing Bti.

Comment 3-5:

Introduction. It is too long, especially descriptions on the mushroom. It should be reduced in half.

Authors’ Response: The Introduction was reduced from 887 to 825 words. Overall, we believe the information about mushroom pests and control tactics provides important background information. We defer to the editors about their opinion if additional text needs to be removed from the Introduction.  

Comment 3-6:

Spores (L144). When were they obtained? Refer to the date.

Authors’ Response: The text was revised to note: “Syringes containing spores of P. columbinus and P. ostreatus suspended in water (Out-Grow Inc., McConnell, Illinois, USA) were used to inoculate sterilized rye berries within glass canning jars (946 ml) or polypropylene grow bags (61 cm × 24.5 cm, l × w; 0.2 μm filter patch) (Figure 2A-C). Spore syringes were obtained in April 2018 and April 2020 for P. columbinus and P. ostreatus, respectively.”

Comment 3-7:

Citation order of figures. Figure 2 is cited before Figure 1 (L208). Number the figures as cited.

Authors’ Response: Figure 1 is actually first mentioned in line 82, which is before the first mention of Figure 2 in line 143.

Comment 3-8:

Figures 1 and 2. These figures may be removed from the manuscript. Without them, the manuscript can be understood by readers.

Authors’ Response: We believe Figures 1 and 2 contribute the context of the manuscript. In particular, we believe Figure 2 provides a useful infographic on the bioassay procedure. We defer to the editors about deleting the figures.

Comment 3-9:

25 cm2 (L213). Do you mean 5 by 5 cm? The figure is more important than the size, since the size can be calculated from the figure, which cannot be known from the size though.

Authors’ Response: The text was revised to note: “After 21 days, yellow sticky cards (5 cm × 5 cm) were affixed to the inside lid of each container to entrap fungus gnats emerging from the substrate (Figure 2C).”

Comment 3-10:

L223-225. Do you mean that the first method was not good to evaluate the efficacy of the agents and that the results could have been biased? If so, Bioassay 1 should be removed from the manuscript.

Authors’ Response: We believe the methods for the bioassays with P. columbinus are valid and do not feel they were biased. To optimize the bioassay methods we felt it was useful to infest with fungus gnats prior to treating the substrate during bioassays with P. ostreatus. We believe the bioassays with P. columbinus and P. ostreatus should both be included in the manuscript since they provide details on two Pleurotus spp.

Comment 3-11:

L354-355. Refer to the other df. F-test is determined with two dfs.

Authors’ Response: The text was revised: “A significant time effect was detected in the weight of substrates treated with Bti, S. feltiae, and water (Within-Subjects, Repeated Measures ANOVA; Figure 5A: F4, 108 = 1308.9; p <0.0001; Figure 5B: F4, 108 = 3335.94; p <0.0001).”

The text was revised: “Substrate weight then consistently decreased over time during both bioassays such that substrate weight at 3 wks after treatment was significantly less compared to the day of treatment with Bti, S. feltiae, and water alone (Figure 5A: F4, 145 = 41.64; p <0.0001; Figure 5B: F4, 145 = 48.26; p <0.0001).”

Comment 3-12:

Figures 3-4. Explain the three panels in the legend of each figure.

Authors’ Response: The captions for Figs 3 and 4 were revised to note: “Three independent bioassays were conducted over time and presented as A-C.”

Comment 3-13:

Figure 5. Errors should be indicated to each mean. If the addition of errors makes the figure messy, consider providing a table.

Authors’ Response: The figure was revised to include standard error bars.

Comment 3-14:

L454. Is it not contradictory with panels B and C in Figure 4?

Authors’ Response: To avoid overstating our results, we felt it was reasonable to note that S. feltiae had “little to no effect” on the emergence of fungus gnats. The sentence was revised to start with “Overall” as a way to indicate that we were commenting about the inclusive results. “Overall, our current study found that straw substrate inoculated with P. columbinus and P. ostreatus and treated with S. feltiae had little to no effect on the emergence of fungus gnats.”